# A Curcumin-Based Oral Gel Has Potential Protective Efficacy against Oral Mucositis: In Vitro Study

**DOI:** 10.3390/jpm14010001

**Published:** 2023-12-19

**Authors:** Majdy Idrees, Omar Kujan

**Affiliations:** Discipline of Oral Pathology, UWA Dental School, The University of Western Australia, 17 Monash Avenue, Nedlands, WA 6009, Australia; majdy.idrees@uwa.edu.au

**Keywords:** oral mucositis, oxidative stress, reactive oxygen species, curcumin oral gel, antioxidant properties, mass spectrometry

## Abstract

Oral mucositis is a common distressing complication of cancer therapy, characterised by painful sores within the oral cavity. Current management options offer limited symptomatic relief. Curcumin, a natural polyphenolic compound with recognised anti-inflammatory and antioxidant properties, has emerged as a potential protective agent against oral mucositis. This study explores the therapeutic potential of curcumin in mitigating the impact of oral mucositis by investigating a commercially available curcumin-based oral gel, PerioGold^®^. Liquid chromatography–tandem mass spectrometry was used to characterise the main constituents of PerioGold^®^. The cytotoxicity of curcumin constituent was investigated in four cell lines: primary oral keratinocytes (HOKs), immortalised oral keratinocytes (OKF6), dysplastic oral keratinocytes (DOKs), and oral squamous cell carcinoma cells (PE/CA-PJ15). Concentrations of hydrogen peroxide were optimised to develop in vitro models that mimic oral mucositis. The inhibitory effects of PerioGold^®^ against the production of reactive oxygen species were assessed using a designated kit. OKF6 cells were the most sensitive to oxidative stress, while PE/CA-PJ15 cells showed the highest resistance. Pretreatment of the investigated cells for 24 h with PerioGold^®^ demonstrated a significant antioxidative effect in all cells based on a dose–response pattern. PerioGold^®^ exhibits clinical potential for protecting against oxidative stress, warranting further individualised clinical investigations.

## 1. Introduction

Head and neck squamous cell carcinoma (HNSCC) makes up over 90% of cancers in the head and neck area, and it is among the ten most common malignancies in humans worldwide [1]. The majority of HNSCCs originate from the mucosal epithelium found in the oral cavity, pharynx, and larynx. Tobacco use, excessive alcohol consumption, or a combination of both are commonly linked to oral cavity and larynx cancers. In contrast, pharynx cancers are increasingly associated with human papillomavirus (HPV) infection, predominantly HPV-16 [1]. In Australia, over 5000 new cases of HNSCCs were documented in 2022, with an average five-year survival rate of 72% [2]. Typically, treatment strategies for HNSCCs combine radiotherapy, chemotherapy, immunotherapy, and surgery, where surgical procedures primarily focus on removing the visible tumour. In contrast, postoperative radiotherapy is employed to prevent potential tumour spread beyond the surgical margins [3].

However, most patients undergoing oncotherapy in the oral cavity regions experience varying degrees of radiotherapy/chemotherapy-induced damage to the oral mucosa [3,4]. These damages are characterised as acute or delayed reactions, where oral mucositis is the most prevalent acute side effect [3]. Approximately 20% to 40% of individuals with solid tumours undergoing chemotherapy experience the onset of mucositis, typically occurring within five to fourteen days after initiating the treatment [3]. Oral mucositis is a highly incapacitating condition, often associated with a range of distressing symptoms, including redness, swelling, severe oral pain, difficulty swallowing (dysphagia), painful swallowing (odynophagia), and discomfort when speaking [3,5]. Additionally, patients with high-grade oral mucositis may experience disruptions in their radiotherapy and chemotherapy treatments, resulting in the resurgence of resistant cell populations and a subsequent compromise in disease control [6]. Whilst the exact pathogenesis of oral mucositis is still to be determined, it is believed to be linked with clonogenic radiation-induced mitotic death of basal keratinocytes and the subsequent release of reactive oxygen species (ROS) by injured cells [5].

Despite its debilitating nature, there is a limited selection of evidence-based therapeutic agents to manage oral mucositis [7,8]. Several natural-product-based therapies, such as calendula, zinc supplementation, and L-glutamine, have been proposed for controlling and limiting signs and symptoms of oral mucositis [8]. Calendula (*Calendula Officinalis*), derived from marigold flowers, has been reported in laboratory studies for its preventive radiation-induced skin toxicity [8,9]. However, clinical studies revealed limited evidence based on conflictive outcomes due to standardisation difficulties [9]. Likewise, zinc supplementation, which supports wound healing, has been proposed as a potential remedy for reducing the severity and duration of oral mucositis [10]. However, excessive zinc intake can lead to adverse effects, and optimal dosing requires careful monitoring [11]. Moreover, a previous meta-analysis showed that zinc supplements did not significantly decrease incidence, severity, or pain intensity among patients with oral mucositis [12]. L-glutamine, an amino acid crucial for mucosal cell growth, has shown contradicting outcomes in terms of mitigating oral mucositis [13]. Moreover, its effectiveness can vary among individuals, and high doses should be avoided due to potential neurological side effects [14].

On the other hand, curcumin, a polyphenol derived from the rhizome of the *Curcuma longa* plant, has gained substantial attention due to its well-known anti-inflammatory, antioxidant, and anticancer properties [15,16,17]. These characteristics position it as a promising remedy for oral mucositis [18]. Curcumin’s therapeutic potential is based on its ability to modulate various cellular and molecular pathways that are fundamental in addressing the pathogenesis of oral mucositis, such as pathways associated with inflammation, oxidative stress, and tissue repair [17,18]. One of these critical mechanisms involves the inhibition of nuclear factor-kappa B (NF-κB), a transcription factor central to inflammation, apoptosis, and cell proliferation [19]. Through the suppression of NF-κB activation, curcumin effectively reduces the expression of pro-inflammatory mediators and helps to alleviate tissue damage [19]. Additionally, curcumin is a scavenger for ROS, countering oxidative stress-induced damage to the oral mucosa. This is achieved through various pathways, including the chelation of metal ions, enhancement of endogenous antioxidants such as superoxide dismutase and catalase, and modulation of cellular signalling by activating the Nrf2/ARE pathway [16,20].

The primary objective of this study was to characterise the main chemical constituents of curcumin within a commercially available product, PerioGold^®^ (Bharma Pty Ltd., Perth, WA, Australia), by carrying out liquid chromatography–tandem mass spectrometry (LC-MS-MS). The second primary objective was to explore the potential antioxidant effects of PerioGold^®^ on oral cells within a controlled in vitro model simulating oral mucositis.

## 2. Material and Methods

### 2.1. Liquid Chromatography–Tandem Mass Spectrometry (LC-MS/MS)

Liquid chromatography–tandem mass spectrometry (LC-MS/MS) was the primary analytical technique for the quantification and identification of target curcumin compounds in samples of a commercially available curcumin-based oral gel (PerioGold^®^, Bharma Pty Ltd., Perth, WA, Australia). The LC-MS/MS system combines the separation abilities of liquid chromatography with the highly sensitive and selective detection capabilities of tandem mass spectrometry, allowing for the precise and simultaneous measurement of multiple analytes. The applied method has been optimised and validated in previous studies using a standard of curcumin (curcumin ≥98.0%, Cat. 08511, Sigma-Aldrich) [21].

In brief, sample preparation involved extraction and sonication for 40 min steps of the investigated samples using DMSO, followed by chromatographic separation on a high-performance liquid chromatography (HPLC) column. The LC system facilitated the elution of analytes, directing them into the mass spectrometer. Compounds were subjected to ionisation, fragmentation, and subsequent detection within the mass spectrometer. Quantification was achieved by comparing analyte peak intensities with those of standard reference compounds. Multiple reaction monitoring (MRM) transitions were utilised for enhanced specificity and sensitivity.

### 2.2. Cell Lines and Culture

In vitro models of four commercially available cell lines were employed in this study to assess the potential antioxidative effects of curcumin-based oral gel in managing cases with oral mucositis. The investigated cell lines were (i) human oral keratinocyte primary cells (HOKs, Cat. 2610, ScienCell, CA, USA) cultured in oral keratinocyte medium (Cat. 2611, ScienCell) and enhanced with 1% penicillin/streptomycin solution (Cat. 0503, ScienCell) and 2% oral keratinocyte growth supplement (Cat. 2652, ScienCell); (ii) immortalised human oral keratinocyte cell line (OKF6, Cat. CRL3397, ATCC, VI, USA) grown in Modified Eagle Medium/Nutrient Mixture F-12 (DMEM/F-12, Cat. 11320033, ThermoFisher, MA, USA) enriched with 10% foetal bovine serum (FBS, Cat. SFBS-F, Bovogen, VIC, Australia), 400 ng/mL hydrocortisone, and 1% antibiotic-antimycotic (ABAM, Cat. 15240062, ThermoFisher); and (iii) dysplastic oral keratinocyte cell line (DOKs, Cat. 94122104, Sigma-Aldrich, MO, USA) and (iv) oral squamous cell carcinoma cell line (PE/CA-PJ15, Cat. 96121230, Sigma-Aldrich), both grown in Advanced Modified Eagle Medium (Advanced DMEM, Cat. 12491015, ThermoFisher), with the addition of 2% Gibco GlutaMAX Supplement (Cat. 35050061, ThermoFisher), 10.3 µM hydrocortisone, and 1% antibiotic-antimycotic (ABAM, Cat. 15240062, ThermoFisher).

### 2.3. Acid-Phosphatase Cytotoxicity Assay (APH) to Identify Suitable Concentrations of the Curcumin-Based Oral Gel

The acid-phosphatase cytotoxicity assay (APH) aimed to assess the cytotoxicity of the curcumin-based oral gel and identify the most suitable concentrations for subsequent analysis. HOKs, OKF6, DOKs, and PE/CA-PJ15 were cultured overnight in 96-well plates with 10,000 cells/well and then exposed to eight different concentrations of curcumin-based oral gel (1 µM, 5 µM, 10 µM, 25 µM, 50 µM, 100 µM, 250 µM, and 500 µM). Each concentration was examined in triplicate for each cell line over five days.

In brief, a complete APH buffer was prepared using 100 mM of sodium acetate, 1.1% Triton X-100, and 2 mg/mL of 4-nitrophenyl phosphate disodium salt hexahydrate (Cat. 71768, Sigma-Aldrich). After removing the culture media from the designated wells, each well was washed twice with 150 µL of phosphate-buffered saline (PBS, Cat. 20012043, ThermoFisher). Subsequently, 150 µL of complete APH buffer was added to each well and incubated for 2 h at 37 °C. The reactions were then terminated by adding 15 µL of 1 N sodium hydroxide (NaOH) to each well. Absorbance at 405 nm was promptly recorded within 10 min using a Sunrise^®^ absorbance reader (Cat. 16039400, Tecan, Grödig, Austria).

### 2.4. In Vitro Model of Oral Mucositis

In this study, we employed an established in vitro model that uses H_2_O_2_, a well-known inducer of oxidative stress, to evaluate curcumin-based oral gel’s potential effectiveness in alleviating oxidative stress’s impact in the context of oral mucositis. This model had been previously developed and validated in other studies [22]. Accordingly, preliminary APH assays were conducted, as explained previously, to determine the investigated cells’ *IC*_50_, which is the concentration of H_2_O_2_ at which 50% of cell viability is inhibited, over a 24 h timeframe. For this purpose, the following concentrations of H_2_O_2_ were assessed: 0 µM (control), 50 µM, 100 µM, 250 µM, 500 µM, 1 mM, and 5 mM. All experiments were conducted in triplicate.

### 2.5. Treatment of the H_2_O_2_-Induced Cells with the Curcumin-Based Oral Gel

The cells were seeded in triplicate in 96-well plates under optimal culture conditions until they reached 70% confluency before being treated with different concentrations of the curcumin-based oral gel for 24 h. The oxidative stress was then induced among the incubated cells using appropriate concentrations of H_2_O_2_ based on individual *IC*_50_
*values* of each cell line. ROS production was measured using a general oxidative stress indicator assay kit (CM-H2DCFDA, Cat. C6827, ThermoFisher) per the manufacturer’s instructions. Indicators of ROS production through 2′,7′-dichlorodihydrofluorescein diacetate (H2DCFDA) were measured using a CLARIOstar fluorescence microplate reader at EM/EX = 492/517 nm. Readings were recorded immediately after the H_2_O_2_ stimulation and every 30 min after that for 5 h.

### 2.6. Statistical Analyses

The statistical analyses, as well as relevant graphical representations, were conducted using GraphPad Prism^®^ (Version 10.0.2, CA, USA). One-way ANOVA and Tukey’s multiple comparison tests evaluated group differences in the APH cytotoxicity assay. The effectiveness of the curcumin-based oral gel against ROS was evaluated using two-way ANOVA and Dunnett’s multiple comparison tests. Dose–response curves using curve-fit nonlinear regression models were individually generated for each cell line to determine the *IC*_50_ concentration of H_2_O_2_. Statistical significance was established at *p* < 0.05.

## 3. Results

### 3.1. Liquid Chromatography–Tandem Mass Spectrometry (LC-MS/MS) Analysis Shows Constituents of the Curcumin-Based Oral Gel and Curcumin Quantity

LC-MS/MS successfully identified 18 chemical constituents within the curcumin-based oral gel, as detailed in Table 1. Apart from curcumin, three components known for their antioxidant properties—diethanolamine, cinnamyl carbonate, and calcium monoselenide—were detected in the investigated samples. Moreover, three constituents were identified with known antimicrobial properties: 1,2,4-triazole, norfloxacin, and dextromethorphan (Table 1). The analysis also provided precise curcumin quantities in the samples, a crucial factor for subsequent experiments, as shown in Figure 1. Accordingly, it has been demonstrated that each gram of PerioGold^®^ includes an average of 216 µg of pure curcumin (Figure 1B). Three separate investigations involving randomly selected samples were performed in triplicate to ensure result consistency and reproducibility.

### 3.2. Acid-Phosphatase Cytotoxicity Assay (APH)

The cytotoxic impact of the curcumin-based oral gel on the viability of the included cell lines over 5 days was evaluated using APH assay. The findings indicate that, when present at concentrations of up to 25 µM, curcumin did not induce cytotoxic effects on HOKs. Consequently, 10 µM and 25 µM concentrations were chosen for further investigations involving this cell line (Figure 2A). In the case of OKF6 and DOKs, the highest curcumin concentration that did not result in cytotoxicity was determined to be 50 µM. Thus, 25 µM and 50 µM concentrations were selected for subsequent experiments, as illustrated in Figure 2B,C. Finally, the APH assay showed that curcumin statistically significantly inhibits the viability of PE/CA-PJ15 at concentrations of 250 µM and more. Therefore, curcumin concentrations of 50 µM and 100 µM were selected for upcoming experiments (Figure 2D).

### 3.3. Identifying the IC_50_ of H_2_O_2_ Concentrations for the In Vitro Oral Mucositis Model

Appropriate *IC*_50_ values based on the concentrations of H_2_O_2_ that inhibit the viability of 50% of the investigated cells were identified in this study to develop the in vitro oral mucositis model. A curve-fit nonlinear regression model was created for each cell line based on seven concentrations of H_2_O_2._ Accordingly, *IC*_50_ values were defined as HOKs (341 µM), OKF6 (134.7 µM), DOKs (532.1 µM), and PE/CA-PJ15 (927.8 µM), as shown in Figure 3.

### 3.4. Efficacy of the Curcumin-Based Oral Gel against Reactive Oxygen Species

In this study, we aimed to assess the effectiveness of curcumin-based oral gel (PerioGold^®^, Bharma, Perth, WA, Australia) in mitigating ROS production in the in vitro oral mucositis models for the investigated cell lines. The results of this investigation showed a statistically significant decrease in ROS production when HOK cells were pretreated for 24 h with curcumin-based oral gel before stimulating them with H_2_O_2_, regardless of the curcumin concentration, as compared with cells exposed to H_2_O_2_ alone, *p* < 0.05 (Figure 4A). This reduction in ROS levels was consistently observed across all investigated time intervals, starting from 0.5 h to 5 h. Furthermore, a dose-dependent effect of curcumin on ROS production was also reported among HOKs; specifically, cell groups pretreated with higher curcumin concentrations (25 µM) exhibited a more pronounced reduction in ROS levels compared with those exposed to lower curcumin doses (10 µM). However, this significant difference in ROS reduction was primarily observed between the 3 h and 4.5 h timepoints, *p* < 0.05 (Figure 4A).

In the case of OKF6 cells, pretreatment with low-dose curcumin (25 µM) for 24 h before stimulation with H_2_O_2_ did not result in a statistically significant reduction in ROS production when compared with groups exposed to H_2_O_2_ alone (*p* > 0.05). Conversely, when OKF6 cells were subjected to high-dose curcumin (50 µM), a statistically significant reduction in ROS production was observed (*p* < 0.05). This reduction commenced from the second hour of exposure and persisted in subsequent time intervals (Figure 4B).

In DOK cells, pretreatment with low curcumin levels (25 µM) failed to yield a statistically significant reduction in ROS production when compared with the groups exposed to H_2_O_2_ alone (*p* > 0.05); this lack of significance persisted until the third hour. Conversely, the application of high-dose curcumin (50 µM) before the H_2_O_2_ stimulation led to a statistically significant reduction in ROS production across all assessed time intervals, *p* < 0.05 (Figure 4C).

The PE/CA-PJ15 cell line displayed a distinctive response to curcumin treatment. Groups pretreated with low doses of curcumin-based oral gel (50 µM) failed to reduce ROS production compared with groups stimulated with H_2_O_2_ alone (Figure 4D). However, high-dose curcumin pretreatment (100 µM) 24 h before stimulation with H_2_O_2_ resulted in a statistically significant reduction in ROS production, and this pattern remained consistent throughout the experiment, *p* < 0.05 (Figure 4D).

## 4. Discussion

Oral mucositis remains a common and distressing side effect of cancer therapy, particularly in patients undergoing chemotherapy and radiation treatments. Curcumin is renowned for its antioxidant and anti-inflammatory properties, making it an attractive candidate for managing oral mucositis [15]. The present study aimed to characterise and evaluate the efficacy of a curcumin-based oral gel, PerioGold^®^ (Bharma Pty Ltd., Perth, WA, Australia), in managing oral mucositis. To achieve this, a comprehensive investigation was conducted using various cell lines and experimental techniques. To the best of our knowledge, this is the first study that assesses the oxidant protective role of curcumin in a commercially available product among a panel of oral cell lines, including dysplastic and carcinogenic cells.

In our investigation, mass spectrometry played a pivotal role in characterising the composition of PerioGold^®^, specifically the presence and quantity of curcumin. The quantification of curcumin content ensures the reliability and consistency of this product for potential therapeutic use. Moreover, LC-MS/MS is often preferred for its superior selectivity and sensitivity, especially in complex matrices or when dealing with low concentrations of specific components [23]. Interestingly, the LC-MS/MS assay identified other parts with antioxidant roles, such as diethanolamine and cinnamyl carbonate [24,25]. Although cinnamyl carbonate is primarily used in industries to contribute to the aroma and scent of products, it is derived from cinnamic acid, an organic compound with various plant sources and established antioxidant activities [25]. The experiments of this study were conducted based on the concentration of curcumin in samples of PerioGold^®^; comprehensively quantifying other components and assessing their role in the developed in vitro model may provide an opportunity to understand the protective mechanism of PerioGold^®^ better and help propose personalised treatment based on individual variables.

As in vitro cytotoxicity tests of therapeutic agents are more sensitive than in vivo tests, they serve as a screening tool to determine the likelihood of any material producing toxicological reactions in patients. Moreover, they play a vital aspect in planning in vitro studies, as they ensure that the observed effects are in a controlled environment and specific to the treatment being investigated, not due to cellular damage or other biological interactions [26,27]. In this study, the sensitivity to curcumin varied from 25µM to 100µM, as highlighted by the differences observed among the investigated cell lines. More importantly, the potential cytotoxicity of the curcumin-based oral gel was assessed in this study over five days rather than 24 or 48 h. This offered several advantages to evaluating cumulative cytotoxic effects that mimic prolonged exposure scenarios, especially when considering that the investigated product was not purely based on curcumin and included other manufacturing components that may exert cytotoxicity.

An integral aspect of this study was the development of an in vitro model that mimics oral mucositis by inducing cells with H_2_O_2_. Such a model is not novel and has been adopted and validated in several previous studies [22,28]. The literature includes several in vitro models that employed various approaches to simulate oxidative stress damage, such as ultraviolet (UV) radiation and ionising radiation [29,30]. However, inducing oxidative stress through H_2_O_2_ offers several advantages over other models. These advantages primarily result from the controlled and targeted nature of H_2_O_2_ exposure compared with UV and ionising radiation, which can be broader and less specific. Moreover, H_2_O_2_ predominantly generates hydroxyl radicals (OH•), a particular type of ROS relevant to oral mucositis pathogenesis [31,32]. UV and ionising radiation, on the other hand, produce a more comprehensive array of ROS, making it challenging to attribute effects solely to a specific ROS type [33].

The distinct responses of different cell lines to H_2_O_2_, as evidenced by the variability in their *IC_50_* values, reveal the complex and cell-type-specific nature of the oral mucositis model. This is seen in the literature, where the range of H_2_O_2_ concentrations to induce oxidative stress spans from 10 µM for cells such as B lymphocytes to over 1000 µM in cancerous cell lines such as uterus squamous cell carcinoma [34,35,36]. This can be attributed to genetic variability, enzymatic defence systems, and cellular redox status [37]. This may also explain the high resistance of DOK PE/CA-PJ15 cell lines to oxidative stress in this study, as they may exhibit alterations in various cellular processes and have developed increased antioxidant systems compared with normal cells due to oncogenic changes [38,39]. More importantly, it reflects heterogeneities among patients undergoing cancer therapy, which necessitate treatment strategies personalised to the individuals’ risk.

The differential protective responses of curcumin against ROS production in the investigated cell lines highlight the cell-specific nature of curcumin’s impact on ROS regulation. Nonetheless, the curcumin-based oral gel significantly reduced ROS production at certain stages and concentrations. Thus, it is worth noting that, due to individual heterogeneities, applying curcumin for therapeutic uses should be based on agents that can stay long in the oral cavity, emphasising the significance of using hydrophobic gel rather than oral rinse. Our results are also consistent with previous studies where curcumin played a significant protective role against ROS in periodontal ligament stem cells [40] and epidermal immortal keratinocyte cells (HaCat) [41].

While this study offers promising insights, there are several limitations to consider. First, this study was conducted in vitro, and in vivo studies are necessary to evaluate the efficacy and safety of curcumin-based treatments in a more complex biological context. Second, the dose-dependent responses observed among different cell lines emphasise the need for precise dosing strategies. Dosing considerations may need further refinement to maximise the therapeutic benefits and minimise potential adverse effects. Finally, translating in vitro findings to clinical applications requires a comprehensive understanding of the pharmacokinetics, pharmacodynamics, and formulation of curcumin-based therapies. Further research is needed to bridge this gap and bring curcumin-based treatments closer to clinical use.

In conclusion, the present study is the first to identify the protective efficacy of a commercially available product based on curcumin against oral mucositis by utilising heterogeneous cell lines where curcumin significantly reduced ROS production. This reported dose–response pattern emphasised the need to utilise mucoadhesive and long-acting formulations. Clinical studies are essential to validate the in vitro findings by comparing them with other current therapies and establishing practical guidelines for managing patients with oral mucositis.

## Figures and Tables

**Figure 1 jpm-14-00001-f001:**
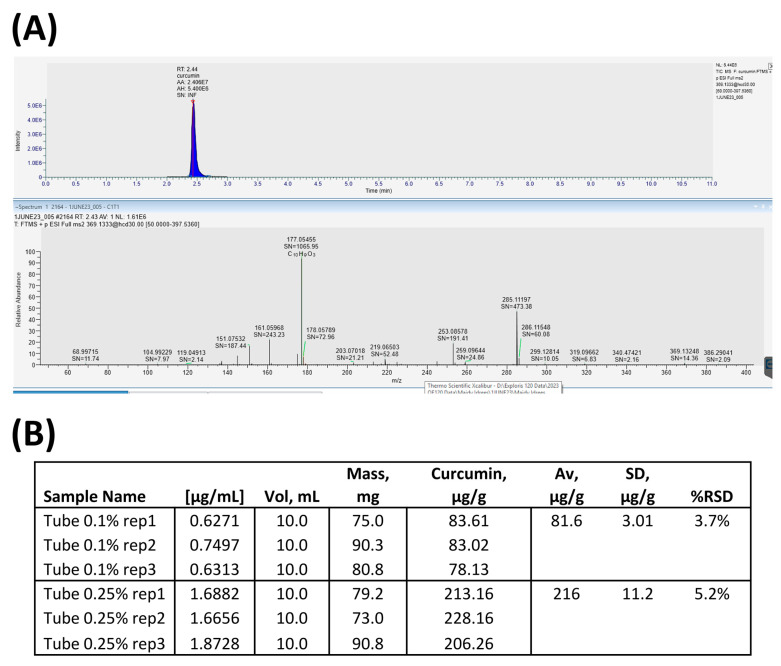
(**A**) Chromatograms showing the intensity and the relative abundance of the constituents of the curcumin-based oral gel; (**B**) quantifications of curcumin in the investigated samples.

**Figure 2 jpm-14-00001-f002:**
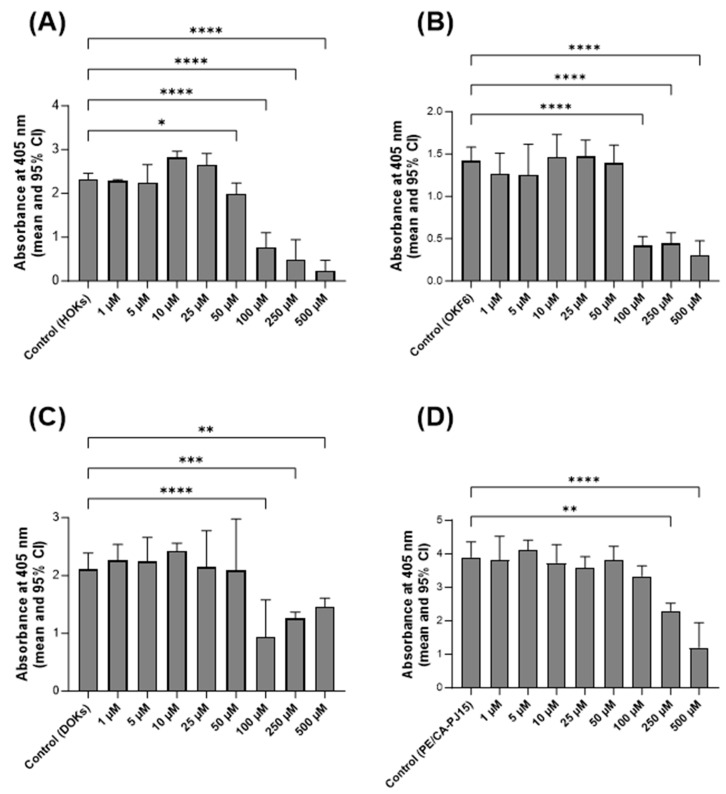
APH to evaluate the cytotoxicity of the curcumin-based oral gel at various concentrations on (**A**) HOKs, (**B**) OKF6, (**C**) DOKs, and (**D**) PE/CA-PJ15. Data are presented as means and 95% confidence intervals (95% CI). Statistical significance is represented as follows: * *p* < 0.05, ** *p* < 0.002, *** *p* < 0.0005, **** *p* < 0.0001.

**Figure 3 jpm-14-00001-f003:**
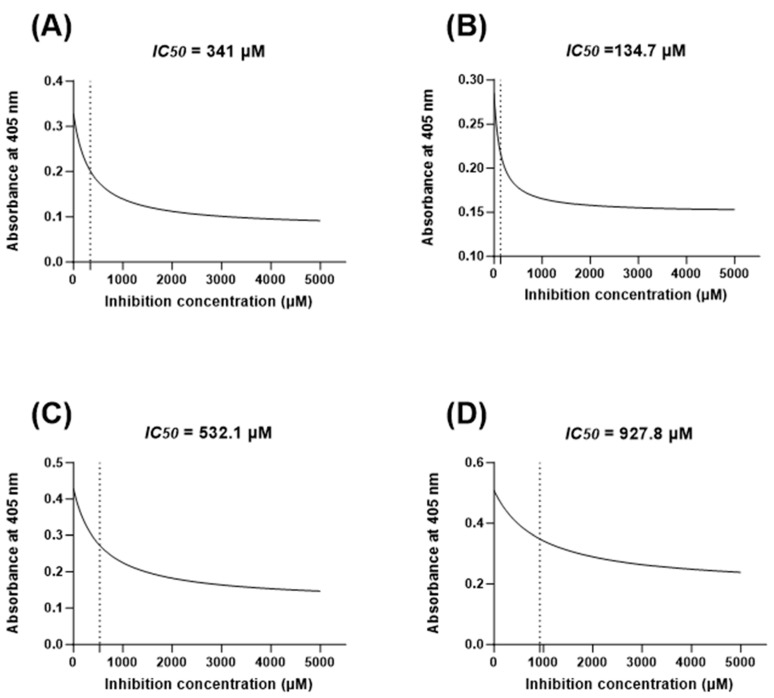
Curve-fit nonlinear regression models (dose-response curves) to determine the concentration of H_2_O_2_ that inhibits 50% of cell viability (*IC*_50_) in 24 h for (**A**) HOKs, (**B**) OKF6, (**C**) DOKs, and (**D**) PE/CA-PJ15.

**Figure 4 jpm-14-00001-f004:**
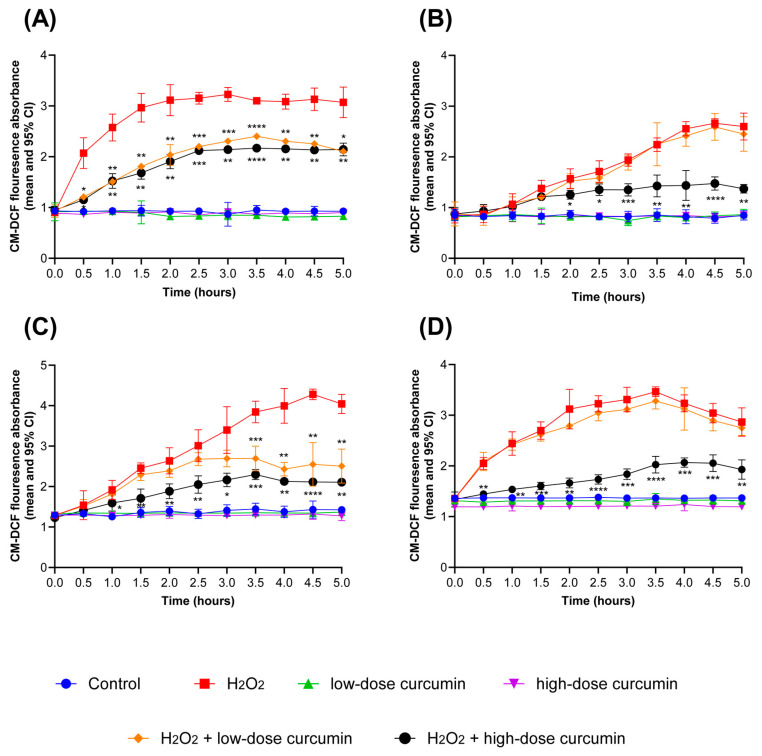
The protective effect of the curcumin-based oral gel (PerioGold^®^, Bharma, Perth, WA, Australia) on ROS production in the model of oxidative stress for (**A**) HOKs, (**B**) OKF6, (**C**) DOKs, and (**D**) PE/CA-PJ15. H_2_O_2_ levels; HOKs (341 µM), OKF6 (134.7 µM), DOKs (532.1 µM), and PE/CA-PJ15 (927.8 µM). Low-dose curcumin; HOKs (10 µM), OKF6 (25 µM), DOKs (25 µM), and PE/CA-PJ15 (50 µM). High-dose curcumin; HOKs (25 µM), OKF6 (50 µM), DOKs (50 µM), and PE/CA-PJ15 (100 µM). Statistical significance values represent the difference in comparison with H_2_O_2_-induced groups. Statistical significance is defined as follows: * *p* < 0.05, ** *p* < 0.002, *** *p* < 0.0005, **** *p* < 0.0001.

**Table 1 jpm-14-00001-t001:** Constituents of the curcumin-based oral gel (PerioGold^®^ Bharma Pty Ltd., Perth, WA, Australia) based on the conducted liquid chromatography–tandem mass spectrometry (LC-MS/MS) analysis.

No.	Molecular Weight	Name	Chemical Formula	Signal Intensity
1	68.99715	1,2,4-Triazole	C_2_H_3_N_3_	11.74
2	104.99229	Diethanolamine	C_4_H_11_NO_2_	7.97
3	119.04913	Calcium monoselenide	CaSe	2.14
4	151.07532	Trifluoromethanesulfonic acid	CF_3_SO_3_H	187.44
5	161.05968	Alpha-aminoadipic acid	C_6_H_11_NO_4_	243023
6	177.05455	Cinnamyl carbonate	C_10_H_9_O_3_	1065.95
7	178.05789	4-Chlorobenzylamine hydrochloride	C_7_H_9_Cl_2_N	72.96
8	203.07018	2,7-Dichloro-1,2,3,8a-tetrahydroquinazoline	C_8_H_8_Cl_2_N_2_	21.21
9	219.06503	2-(2,6-Dichlorophenyl)acetohydrazide	C_8_H_8_Cl_2_N_2_O	52.48
10	253.08578	Diethyl 2-bromobutanedioate	C_8_H_13_BrO_4_	191.41
11	259.09644	Methyl 4-(bromomethyl)-2-methoxybenzoate	C_10_H_11_BrO_3_	24.86
12	285.11197	1-(Bromomethyl)-4-[(difluoromethyl)sulfonyl]benzene	C_8_H_7_BrF_2_O_2S_	473.38
13	286.11548	4-Bromo-5,7-difluoro-2-propylquinoline	C_12_H_10_BrF_2_N	60.08
14	299.12814	5-Amino-3-chloro-2-octoxybenzoic acid	C_15_H_22_ClNO_3_	10.05
15	319.09662	Norfloxacin	C_16_H_18_FN_3_O_3_	6.83
16	340.47421	Behenic acid	C_22_H_44_O_2_	2.16
17	369.13248	Dextromethorphan	C_18_H_25_NO	14.36
18	386.29041	1-O-Sinapoyl-beta-D-glucose	C_17_H_22_O_10_	2.09

## Data Availability

The data supporting this study’s findings are available on request from the corresponding author. The data are not publicly available due to privacy.

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
