# Peer review of "A Curcumin-Based Oral Gel Has Potential Protective Efficacy against Oral Mucositis: In Vitro Study"

_jpm, 2023, doi:10.3390/jpm14010001_

Round 1
Reviewer 1 Report
Comments and Suggestions for Authors
Dear authors, the topic is very interesting, but I suggest some insights and tips to make your manuscript good for publication:
line 28 : Head and neck squamous cell carcinomas (HNSCCs) , you need to go deeper it is too short one line
line 38: explain better what mucositis is and what it can be caused by other than radiotherapy.
line 254: expand on what curcumin is, cite articles that talk about, cite and explain curcumin.
divide the limitations of the study from the discussion and also the conclusions.
Discussion, limitations and conclusions should be 3 separate sections.
Author Response
Great thanks for your time in reviewing our manuscript and providing helpful comments to improve the clarity of our work.
line 28 : Head and neck squamous cell carcinomas (HNSCCs) , you need to go deeper it is too short one line
Thank you. This has been expanded.
line 38: explain better what mucositis is and what it can be caused by other than radiotherapy.
Thanks. Valid point. We revised the paragraphs to provide abetter description of mucositis.
line 254: expand on what curcumin is, cite articles that talk about, cite and explain curcumin.
Thanks. This has been described in the introduction and I’m afraid of repetition if I add an explanatory paragraph on Curcumin at the beginning of the discussion section.
divide the limitations of the study from the discussion and also the conclusions.
Discussion, limitations and conclusions should be 3 separate sections.
Thanks. We value your comment, and this has been done.
Reviewer 2 Report
Comments and Suggestions for Authors
The study titled "A Curcumin-Based Oral Gel Has a Potential Protective Efficacy Against Oral Mucositis: In Vitro Study" investigates the therapeutic potential of a commercially available curcumin-based oral gel, PerioGold®, in treating oral mucositis, a common side effect of cancer therapy. The manuscript is well-written, with a logical flow and a clear presentation of the study's objectives, methodology, results, and implications. The study adds valuable information to the field of personalized medicine, particularly in the management of oral mucositis, a common complication in cancer patients. The findings are promising, suggesting that PerioGold® could potentially offer a new therapeutic avenue for managing oral mucositis. However, the transition from in-vitro to in-vivo studies and eventually to clinical trials is necessary to fully understand the efficacy and safety of curcumin-based oral gel in real-world settings. There are some issues and questions that should be addressed in the study.
1- Provide more details on the composition of PerioGold® to understand the role of other constituents in its therapeutic efficacy.
2- Page 1 Line 31 “Typically, treatment strategies for HNSCCs combine radiotherapy and surgery, where surgical procedures primarily focus on removing the visible tumour” Please also include chemotherapy and immunotherapy as treatment options”
3- Page 2 Line 63 “ On the other hand, curcumin, a polyphenol derived from the rhizome of the Curcuma longa plant, has gained substantial attention due to its well-known anti-inflammatory, antioxidant, and anti-cancer properties” Please add recent references for this (https://doi.org/10.1080/10408398.2021.1976721)
4- The study does not compare the efficacy of PerioGold® with existing standard treatments for oral mucositis. Such comparisons could help position the gel within the current therapeutic landscape.
Comments on the Quality of English Language
Moderate editing of English language required
Author Response
We highly value the comments and feedback regarding our manuscript, and we thank you for your time and intellectual input.
- Provide more details on the composition of PerioGold® to understand the role of other constituents in its therapeutic efficacy.
Thank you. This is has already outlined in Table 1.
- Page 1 Line 31 “Typically, treatment strategies for HNSCCs combine radiotherapy and surgery, where surgical procedures primarily focus on removing the visible tumour” Please also include chemotherapy and immunotherapy as treatment options”
Thanks. This has been revised to include all types of head and neck cancer therapies.
- Page 2 Line 63 “ On the other hand, curcumin, a polyphenol derived from the rhizome of the Curcuma longa plant, has gained substantial attention due to its well-known anti-inflammatory, antioxidant, and anti-cancer properties” Please add recent references for this (https://doi.org/10.1080/10408398.2021.1976721)
Thanks. This has been added.
- The study does not compare the efficacy of PerioGold® with existing standard treatments for oral mucositis. Such comparisons could help position the gel within the current therapeutic landscape.
Thanks. In this study we opted not to compare the curcumin-based gel with existing topical oral mucositis therapies to avoid conflict of interests. However, we added this point as a future research direction.
Round 2
Reviewer 1 Report
Comments and Suggestions for Authors
Dear authors, I have re-read and re-analysed the manuscript.
You have greatly improved your manuscript by making it more interesting.
I do not ask you for any further changes
Congratulations on your work!